# Expansive Latent Planning for Sparse Reward Offline Reinforcement Learning

**Robert Gieselmann**
KTH Royal Institute of Technology
Sweden
robgie@kth.se

**Florian T. Pokorny**
KTH Royal Institute of Technology
Sweden
fpokorny@kth.se

**Abstract:** Sampling-based motion planning algorithms excel at searching global solution paths in geometrically complex settings. However, classical approaches, such as RRT, are difficult to scale beyond low-dimensional search spaces and rely on privileged knowledge e.g. about collision detection and underlying state distances. In this work, we take a step towards the integration of sampling-based planning into the reinforcement learning framework to solve sparse-reward control tasks from high-dimensional inputs. Our method, called VELAP, determines sequences of waypoints through sampling-based exploration in a learned state embedding. Unlike other sampling-based techniques, we iteratively expand a tree-based memory of visited latent areas, which is leveraged to explore a larger portion of the latent space for a given number of search iterations. We demonstrate state-of-the-art results in learning control from offline data in the context of vision-based manipulation under sparse reward feedback. Our method extends the set of available planning tools in model-based reinforcement learning by adding a latent planner that searches globally for feasible paths instead of being bound to a fixed prediction horizon.

**Keywords:** Reinforcement Learning, Planning, Robot Manipulation

## 1 Introduction

The acquisition of complex motor skills from raw sensory observations presents one of the main goals of robot learning. Reinforcement learning (RL) [1] provides a generic framework to obtain such decision-making policies through the interaction with an environment. Model-based RL [2] has recently gained much attention due to benefits in terms of sample-efficiency and robustness in long-horizon scenarios. To address the issue of short-sighted decisions, model-based agents are often equipped with planning methods. However, effective planning with high-dimensional inputs, such as video data, is often challenging due to the increased complexity of the search space and the difficulty in generating accurate long-term predictions. Consequently, a growing body of research has explored the utilization of representation learning to simplify the decision-making problem by mapping it to an abstract and lower-dimensional latent state space [3, 4, 5, 6, 7].

The model-based reinforcement learning (RL) literature has investigated various planning methods in latent spaces, encompassing zero-order shooting-based approaches such as the Cross-Entropy Method (CEM) [8, 3] and Model-Predictive Path Integral (MPPI) [9, 10, 11, 7], first-order gradient-based optimization [12, 4], and more recently, trajectory collocation using second-order solvers [6]. Despite this methodological diversity, the majority of existing tools primarily facilitate local optimization within a fixed prediction horizon. Even with guidance from value heuristics, such as the one proposed in [7], local minima may still impede progress, particularly when estimating the optimal value function is difficult due to sparse reward feedback or limited training data. This paper argues that planning in latent state spaces can benefit from more global exploration strategies that seek solutions beyond a fixed prediction horizon to avoid convergence to local minima.

7th Conference on Robot Learning (CoRL 2023), Atlanta, USA.

The limitations observed in existing methods raise the need for more sophisticated planning strategies that can seamlessly integrate with learned state and dynamics models. Sampling-based motion planning [13], provides a diverse range of algorithms for finding global paths between states in continuous and geometrically-complex environments. Recent works by [5, 14, 15] have proposed modifications of sampling-based planners in latent spaces. However, these approaches either rely on expert data or are not directly applicable to the reward-based learning

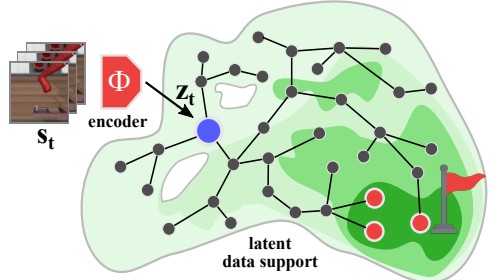

Figure 1: Our method grows a search tree in the latent space to globally explore reward-maximizing paths (blue:start, red:goal nodes, green: estimated values).

setting. This paper explores the integration of sampling-based planning techniques in learned latent spaces, providing new avenues for model-based reinforcement learning. Specifically, we focus on the challenging scenario of offline RL [16], which is characterized by the amplified effects of value approximation errors [17]. Further, it allows us to better study the performance of planning in isolation by disentangling training and data collection. We introduce Value-guided Expansive Latent Planning (VELAP), which combines a sampling-based planning module with a suitable state embedding. Similar to [15], our search tree serves as a state memory and it used to guide the exploration towards undiscovered areas within the data support. Moreover, we leverage value heuristics obtained through temporal difference learning to accelerate the discovery of high-valued states. We present a comprehensive benchmark evaluation focusing on vision-based control. For this purpose, we adapt the robot manipulator control environments from the meta-world benchmark suite [18]. Our experiments reveal that VELAP surpasses existing approaches by a significant margin in terms of episode success rate. We attribute this performance gain to its ability to overcome local value optima through global exploration, in contrast to the prevalent approach of optimizing over a fixed horizons.

## 2   Related Work

Learning to control from visual input is becoming increasingly popular due to the wide availability of inexpensive sensors and the generic representational format of images. A considerable number of work uses deep generative models [19] to generate future images and plan actions via model predictive control [20, 21, 22, 23, 24]. These methods allow visual inspection of paths by humans, but this is accompanied by the difficulty of generating and evaluating high-resolution video sequences of many future steps. In this work, we instead follow the latent planning paradigm, which bypasses the need to synthesize high-dimensional samples and enables farsighted search with lower computational costs. Several works learn image distance metrics using unsupervised learning [25, 5] or RL [26, 27] to build an environment map that can be searched to generate visual paths. Map-based approaches have shown impressive results for navigation in static environments, but often do not scale well to complex settings with object interaction.

The work in [3] presents a model-based RL agent which uses a recurrent state space model to map image observations to a lower-dimensional latent space. Time-discrete state-action trajectories are optimized using the Cross-Entropy Method (CEM) [8] with a learned latent dynamics model. In a follow-up work, [4] uses a differentiable latent planner to efficiently learn behaviors by propagating analytic value gradients back in time. Similarly, [12, 28] implement a differentiable planner that optimizes latent trajectories via gradient descent. [29] trains a goal-conditioned RL agent which generates future subgoal states through trajectory optimization within the latent space of a Variational Autoencoder (VAE) [30]. [6] introduces the concept of collocation to model-based RL for visual robot manipulation and optimizes state-action sequences directly with a second-order optimizer. [10] solve dexterous manipulation tasks through latent trajectory optimization using a reward-weighted adaptation of MPPI [9]. Similarly, the method in [7] uses value estimates as cost within an MPPI-based latent policy. The line of work in [31, 32, 33] combines model-based RL

with Monte Carlo Tree Search (MCTS) [34] for long-horizon decision-making in vision-based tasks such as playing Atari games. While MCTS is mainly designed for discrete settings, [35] presents an adaptation for continuous action spaces where the search and the policy improvement are based on action sampling.

To address the offline RL setting, [36] modifies the MPPI agent in [10]. Their approach ensures adherence to the data support by sampling actions from an imitation learning policy, thus addressing the issue of out-of-distribution actions [17]. Similar, [11] presents a model-based agent for online and offline RL which uses MPPI to maximize the expected return of imagined trajectories while being guided by a learned policy. [37] designed a hierarchical agent for sparse reward manipulation tasks. A VAE-based manager policy predicts subgoal states which a worker policy must achieve within a fixed contingent of steps. In [38], a RRT-like [39] latent planner is presented for planning from high-dimensional data. Compared to ours, their method relies on collision checking data and is not designed for the more general reward-based setting. [14] leverages play data obtained from a human operator to train a task-conditioned policy which guides a tree search in a learned latent space. Our method is most related to the one recently presented in [15]. The authors introduce a sampling-based latent planner similar to the classical Expansive Space Trees (EST) algorithm [40]. Nevertheless, notable differences arise in terms of the problem types we tackle, resulting in the adoption of distinct sets of tools. While their approach is confined to goal-reaching navigation tasks, our method accommodates more general task specifications by leveraging sparse rewards. To accomplish this, we employ value-based reinforcement learning to (a) jointly optimize representations for planning alongside the control policy, (b) integrate learned heuristics for node and action selection during planning, and (c) identify suitable goals based on value estimates. These advancements significantly enhance the capabilities and versatility of our method, surpassing the scope of the previous work.

## 3 Preliminaries

**MDPs and Offline RL** A Markov decision process (MDP) is defined by a tuple $\mathcal{M} = (\mathcal{S}, \mathcal{A}, \mathcal{P}, r, \gamma)$, where $\mathcal{S}$ and $\mathcal{A}$ are state and action spaces, $\mathcal{P}(s'|s, a)$ are state dynamics, $r(s, a)$ is a scalar reward function, and $\gamma$ is a discount factor. The goal of reinforcement learning [1] (RL) is to find a policy $\pi(a|s)$ that maximizes the expected discounted future reward $R[\tau]$ over all trajectories $\tau$ given an initial state distribution $p_0$ and induced by $\pi$, i.e., to optimize $\mathbb{E}_\pi[R[\tau]]$. The problem of offline RL [16] arises when training from a fixed dataset $\mathcal{D}$ consisting of trajectories generated by a behavior policy $\pi_\beta$. Due to the limited coverage of $\mathcal{D}$ across the state-action space, effectively addressing the adverse consequences of poor approximations outside the data support becomes crucial in the development of offline RL methods [17].

**Hindsight data relabeling** Relabeling data has emerged as a popular technique in goal-conditioned off-policy RL [41, 42, 43, 44, 45] for the purpose of enhancing training efficiency. The underlying idea behind hindsight relabeling is to transform unsuccessful trajectories into successful ones by retrospectively modifying their goals [41]. This approach extends to offline trajectory datasets, where relabeling can be used to synthesize experiences for learning state-reaching behavior [46, 47]. Failed transitions are relabeled by designating the subsequent state as the desired goal and adjusting the corresponding reward accordingly. A connection between hindsight relabeling and contrastive learning was recently discussed in [48].

**Sampling-based motion planning** Sampling-based motion planners [13] compute feasible paths connecting two points in a robot configuration space. At the core, these methods explore and construct a graphical representation of the continuous search space. The rapidly-exploring random tree (RRT) [39] is a widely used single-query planner, particularly suitable for scenarios with varying environments. It incrementally expands a tree structure by alternately sampling collision-free states from the robot's configuration space and attempting to connect these to the nearest neighbor in the tree. Once a node reaches the vicinity of the target, a possible solution path is given by backtracing

to the root of the tree. Instead of sampling states from the configuration space, the expansive space trees (EST) planner [40] generates new states by expanding existing tree nodes using randomly-sampled actions.

# 4  Value-guided Expansive Latent Trees

In this section, we detail the elements that comprise VELAP, our proposed offline RL planning agent.

**Problem definition**  We are interested in solving sparse reward continuous control tasks from high-dimensional inputs. For this purpose, we choose the example of visual control for a state space $\mathcal{S}$ and action space $\mathcal{A} = \mathbb{R}^{d_{\text{action}}}$. $\mathcal{S} = \mathbb{R}^{W \times H \times C \times N}$ describes sequential image data where W is the image width, H the height, C the channel dimension and N the number of frames. Note that we employ the MDP formulation, hence assume that states $s \in \mathcal{S}$ are informative to predict the distribution of future states. A sparse binary reward $r : \mathcal{S} \times \mathcal{A} \rightarrow \{0, 1\}$ is designed which provides a positive signal only when reaching the final goal for which we terminate the episode. For training, we use an offline dataset $\mathcal{D}$ consisting of recorded transitions obtained from a sub-optimal policy.

**Components**  To tackle the specified problem, we propose a model-based RL agent that incorporates a tree-based search, inspired by ESTs [40], within a learned representation space. Our preference for EST over an RRT-based approach [39] is driven by the reasoning that the ex-

$$
\begin{aligned}
\text{State encoder: } & \phi : \mathcal{S} \rightarrow \mathcal{Z} \\
\text{Dynamics: } & h : \mathcal{Z} \times \mathcal{A} \rightarrow \mathcal{Z} \quad\quad (1) \\
\text{Action model: } & g : \mathcal{Z} \times \mathbb{R}^m \rightarrow \mathcal{A} \\
\text{Local policy: } & \pi^l : \mathcal{Z} \times \mathcal{Z} \rightarrow \mathcal{A} \quad Q^l : \mathcal{Z} \times \mathcal{Z} \times \mathcal{A} \rightarrow \mathbb{R} \\
\text{Global policy: } & \pi^g : \mathcal{Z} \rightarrow \mathcal{A} \quad\quad Q^g : \mathcal{Z} \times \mathcal{A} \rightarrow \mathbb{R}
\end{aligned}
$$

pansion step in ESTs eliminates the need for a global state sampler. It should be noted that learning such generative models can be challenging, as they require high-fidelity predictions to prevent negative assessment of out-of-distribution samples. Our approach involves several key components outlined in Eq. 1. The encoder $\phi$ maps input states to latent encodings, while the dynamics model $h$ predicts future latent states based on actions, serving as a tool for expanding the search tree during planning. A local policy $\pi^l$ is trained to navigate between neighboring states in the tree. The global policy $\pi^g$ determines optimal actions with respect to our task goal. During planning, we will use $Q^l$ to derive a distance proxy between states and $Q^g$ to estimate the remaining number of steps to the goal. Among various actor-critic offline RL methods available, we select TD3-BC [49] due to its robustness and ease of implementation. To improve the predictions of $Q^l$ and measure value uncertainty, we employ an ensemble of $n_{\text{ens}}$ Q-heads $\{Q^l_1, ..Q^l_{n_{\text{ens}}}\}$ similar to [50] (see App. B). For the following, we use $k$ to denote the $k$-th ensemble member and define $Q^{i,j}_{\min} := \min\{Q^l_k(z_i, z_j, \pi^l(z_i, z_j))\}^{n_{\text{ens}}}_{k=1}$ as the minimum and $Q^{i,j}_{\text{std}} := \text{std}(\{Q^l_k(z_i, z_j, \pi^l(z_i, z_j))\}^{n_{\text{ens}}}_{k=1})$ as the standard deviation of the ensemble predictions between two states $z_i$ and $z_j$ with respect to $\pi^l$. Finally, a conditional generative model, representing our action model $g$, enables sampling actions from the state-conditioned action distribution. We use $\mathbb{R}^m$ to denote the input noise used during the generation process.

**Alignment of representation and planner**  To achieve long-horizon planning and control in $\mathcal{Z}$, we seek a state representation which favors accurate learning of dynamics in order to generate valid future waypoint states over many time steps. Secondly, the state encoding should facilitate the optimization of our value functions and control policies. Existing model-based RL approaches often rely on surrogate metrics for model learning, such as mean-squared prediction error or pixel-wise reconstruction. These metrics do not ensure alignment with actual control performance, leading to a mismatch between the environment model and the planner [51], which can adversely affect the controller's performance. To address the challenge of long-horizon predictions, we optimize our state encoder $\phi$ together with the latent dynamics $h$. In addition, we facilitate the approximation of the local and global value functions by training their models jointly with the encoding. Our model training objective $\mathcal{L}_{\text{model}}$ is shown in Eq. 2. Here, $\mathcal{L}_{Q^l_k}$ represents the temporal difference (TD) loss

for training $Q_k^l$, $\mathcal{L}_{Q^g}$ corresponds to the TD loss for training $Q^g$, and $\mathcal{L}_h$ denotes the loss function for the dynamics model $h$. The hyperparameters $c_0$ and $c_1$ act as weighting factors.

$$\mathcal{L}_{\text{model}} = \frac{1}{n_{\text{ens}}} \sum_k \mathcal{L}_{Q_k^l} + c_0 \cdot \mathcal{L}_{Q^g} + c_1 \cdot \mathcal{L}_h \tag{2}$$

$$\mathcal{L}_{Q_k^l} = \mathbb{E}_{\mathcal{D}'}[(Q_k^l(z_t, z_*, a_t) - (r_t + \gamma Q_{\min}^{t+1,*}))^2] \tag{3}$$

$$\mathcal{L}_{Q^g} = \mathbb{E}_{\mathcal{D}}[(Q^g(z_t, a_t) - (r_t + \gamma Q^g(z_{t+1}, \pi^g(z_{t+1}))))^2] \tag{4}$$

In accordance to the standard TD3-BC training objective, we simultaneously optimize the corresponding policies $\pi^l$ and $\pi^g$ in $\mathcal{L}_{\pi^l}$ and $\mathcal{L}_{\pi^g}$ (Eq. 5). Note that this step is done by alternating between optimizing $\mathcal{L}_{\text{model}}$ and policy improvement while the encoder parameters are kept fixed during the policy update[1].

$$
\begin{aligned}
\mathcal{L}_{\pi^l} &= \mathbb{E}_{\mathcal{D}'}[-Q_{\min}^{t,*}] + c_2 \cdot \mathbb{E}_{\mathcal{D}'}[(\pi^l(z_t, z_*) - a_t)^2] \\
\mathcal{L}_{\pi^g} &= \mathbb{E}_{\mathcal{D}}[-Q^g(z_t, \pi^g(z_t))] + c_3 \cdot \mathbb{E}_{\mathcal{D}}[(\pi^g(z_t) - a_t)^2]
\end{aligned}
\tag{5}
$$

To provide data for training the local policy and value functions $\pi^l$ and $Q_k^l$, we synthesize a dataset of state-reaching experiences $\mathcal{D}'$ by relabeling the transitions in $\mathcal{D}$. More specifically, we achieve this using hindsight goal relabeling [46, 47] to sample goals $s_* \in \mathcal{S}$ and use a binary reward to indicate success (see App. B.4). For training the dynamics model, we use the contrastive loss presented in [53]. In practice, we found this approach to work better in maintaining accurate long-term predictions compared to a standard mean-squared error objective (App. E.3).

**Tree expansion**  Our aim is to solve the RL decision-making problem by searching the latent state space for the shortest connection towards valid goal states. Similar to [15], we follow the concept of EST planners [40] which iteratively expand the current set of nodes through action sampling. The tree $\mathcal{T}=(\mathcal{V},\mathcal{E})$ can be seen as a growing memory of latent nodes $\mathcal{V} \subset \mathcal{Z}$ and transitions $\mathcal{E} \subset \mathcal{Z} \times \mathcal{Z}$. The core mechanism behind our expansion strategy is summarized in Alg. 1. We first initialize $\mathcal{T} = (\mathcal{V}=\{z_{\text{init}}\}, \mathcal{E}=\emptyset)$ where $z_{\text{init}} \in \mathcal{Z}$ is the latent encoding of the current state $s_{\text{init}} \in \mathcal{S}$ obtained from $\phi$. For $n_{\text{iter}}$ steps, a node $z_{\text{expand}}$ is drawn using a categorical distribution $P_{\text{node}}$ defined over $\mathcal{V}$. Starting from $z_{\text{expand}}$, the dynamics $h$ rolls out a short $n_{\text{sim}}$-step

---

**Algorithm 1** Node sampling and tree expansion

1: Given: $z_{\text{init}}, n_{\text{iter}}, n_{\text{sim}}, g, h, \pi^g, Q^g, Q_k^l, \pi^l$
2: Initialize: $\mathcal{V} \leftarrow \{z_{\text{init}}\}, \mathcal{E} \leftarrow \emptyset$
3: **for** $n_{\text{iter}}$ steps **do**
4:     Sample node $z_{\text{exp}}$ from $\mathcal{V}$ given $P_{\text{node}}(\mathcal{V})$
5:     $z_{\text{new}} \leftarrow z_{\text{exp}}$
6:     Simulate forward using dynamics for $n_{\text{sim}}$ steps
7:     **for** $n_{\text{sim}}$ steps **do**
8:         Sample action $a \sim g(.|z_{\text{new}})$ (or $a = \pi^g(z_{\text{new}})$)
9:         $z_{\text{new}} \leftarrow h(z_{\text{new}}, a)$
10:     **end for**
11:     Reject node if too close to existing one in the tree or
12:     if the value uncertainty is too high
13:     **if** $Q_{\min}^{\text{exp,new}} > \tau_{\text{discard}}^{\text{low}}$ and $Q_{\text{std}}^{\text{exp,new}} < \tau_{\text{discard}}^{\text{std}}$ **then**
14:         **if** $\max\{Q_{\min}^{i,\text{new}}|z_i \in \mathcal{V}\} < \tau_{\text{discard}}^{\text{high}}$ **then**
15:             Add new node to tree
16:             $V \leftarrow V \cup \{z_{\text{new}}\}; E \leftarrow E \cup \{z_{\text{exp} \to \text{new}}\}$
17:         **end if**
18:     **end if**
19: **end for**

---

state sequence given actions drawn from our generative model $g$ (or $\pi^g$). Since $Q_k^l$ estimates the return for reaching towards a particular node under sparse binary rewards, a temporal distance proxy is given by $\log_\gamma Q_k^l$. To account for value approximation errors [17], we will use the minimum value among the ensembles predictions to compute a conservative distance estimate. After every $n_{\text{sim}}$-step expansion with $h$, we determine if the transition from $z_{\text{exp}}$ to $z_{\text{new}}$ is feasible by checking if $Q_{\min}^{\text{exp,new}}$ is above a threshold $\tau_{\text{discard}}^{\text{low}}$. If it lies below this threshold, we discard $z_{\text{new}}$. Secondly, we also reject it if the corresponding value of $Q_{\text{std}}^{\text{exp,new}}$ is above a threshold $\tau_{\text{discard}}^{\text{std}}$. The purpose of this second rejection step is filter states in which the epistemic uncertainty, i.e. model uncertainty, is high and thereby avoid the evaluation of high-uncertainty areas, for example outside the support of the latent

---

[1]We optimize the state representation during the critic update instead of the policy improvement step as motivated by the empirical analysis in [52].

data distribution. Lastly, we determine if the newly generated node is sufficiently novel given the existing ones in $\mathcal{T}$ and discard it otherwise. This sparsification step avoids redundant tree nodes and is important to keep the computations at a moderate level. More specifically, we discard $z_{\text{new}}$ if $\max\{Q_{\min}^{i,\text{new}}|z_i \in \mathcal{V}\}$ is above a threshold $\tau_{\text{discard}}^{\text{high}}$. In other words, we find the closest neighbor $z_{\text{neigh}}$ in the tree and reject $z_{\text{new}}$ if there already exists a node which can transition to it within few steps. If $z_{\text{new}}$ passes the previous stages, it is added to $\mathcal{T}$, i.e. $\mathcal{V}{\leftarrow}\mathcal{V} \cup \{z_{\text{new}}\}$ and $\mathcal{E}{\leftarrow}\mathcal{E} \cup \{z_{\text{exp}\rightarrow\text{new}}\}$. For more details on the expansion step see App. B.6.

**Node sampling** So far we haven't defined the node sampling distribution $P_{\text{node}}$. To achieve fast and task-oriented exploration, we combine two sampling heuristics based on (a) the inverse number of neighbors around each node and (b) the state-action value $Q^g$. (a) leads to quick exploration of undiscovered latent states, while (b) drives the planner towards high-valued areas. For

$$P_{\text{sparse}}(z_i) = \frac{e^{-n_i^{\text{neigh}}/T_{\text{sparsity}}}}{\sum_{z_j \in \mathcal{V}} e^{-n_j^{\text{neigh}}/T_{\text{sparsity}}}} \quad (6)$$

$$P_{\text{value}}(z_i) = \frac{e^{Q_i^g/T_{\text{value}}}}{\sum_{z_j \in \mathcal{V}} e^{Q_j^g/T_{\text{value}}}} \quad (7)$$

both parts, we use exponential weighting as shown in Eq. 6 and 7. In this regard, $n_i^{\text{neigh}}$ corresponds to the number of incoming neighbors for a node ($\mathcal{V}_{\rightarrow i}^{\text{neigh}}$). We compose $P_{\text{node}}$ by sampling according to $P_{\text{sparse}}$ with probability $p^{\text{sparse}}$ and from $P_{\text{value}}$ with $p^{\text{value}}$ (otherwise random uniform).

**Action sampling** The model $g$ mimics the state-dependent action distribution of our data and is represented by a conditional VAE [30]. Sampling actions from $g$ reduces the evaluation of undesired state-actions pairs which were not observed in $\mathcal{D}$. To help our planner discover task-relevant areas quicker, we further predict actions using $\pi^g$ with a probability $p^{\text{policy}}$.

**Planning objective and control** Our planner builds a sparse tree representation in the latent space whose expansion is guided by value and sparsity heuristics. To choose the best path in $\mathcal{T}$, we must define an objective that ranks all explored paths. In practice, we first identify all nodes for which the value of $Q^g$ surpasses a threshold $\tau_{\text{goal}}$ and gather the associated paths from the root $z_{\text{init}}$ in a set $G$. Among the elements in $G$, we then choose the path $g^*$ which has the shortest temporal length using $Q_{\min}^{i,j}$ to derive a distance proxy between subsequent nodes (Eq. 8). If $G = \emptyset$, we simply pick the path that contains the node with the highest value of $Q^g$.

$$g^* = \underset{g \in \mathcal{G}}{\arg\min}\, c(g) \qquad \text{with} \qquad c(g) = \sum_{(i,j)\in\mathcal{E}_g} \log_\gamma Q_{\min}^{i,j} \qquad (8)$$

To use our planner for control, we embed it into a model-predictive control loop (Alg. 2). The controller queries our planner every $n_{\text{replan}}$ steps and uses the local policy $\pi^l$ to navigate between nodes in the planned sequences of latent states. If close enough, the controller switches to the next waypoint, which we determine by checking the value of $Q_{\min}^{i,j}$ against a threshold $\tau_{\text{wp}}$.

## 5 Experiments

### 5.1 Empirical evaluation in simulation

**Baselines** To assess the effectiveness of VELAP, we measured its performance against the following baselines. *Behavioral cloning* (**BC**), a simple but often successful method that imitates the behavior policy using a supervised learning objective. We evaluated a second version of this approach only trained on the subset of successful trajectories ($\mathcal{D}^*$). **TD3-BC** [49], an adaptation of the Twin Delayed DDPG algorithm [54] which circumvents the effect value overestimation by adding an imitation objective to the policy update. **IQL** [55], a state-of-the-art offline RL baseline. **MPPI** [9] provides the base planning algorithm in various model-based RL methods ([10, 7]). We consider an implementation for the offline learning setup which uses TD3-BC within the cost update during optimization. **MBOP** [36], a model-based agent which uses an imitation learning policy to bias the action sampling in MPPI. **IRIS** [37], an offline RL method particularly designed for sparse reward settings. It uses a hierarchical decomposition of the policy for which a manager predicts feasible subgoals given future candidate states (n-step horizon) sampled from a generative model (cVAE)

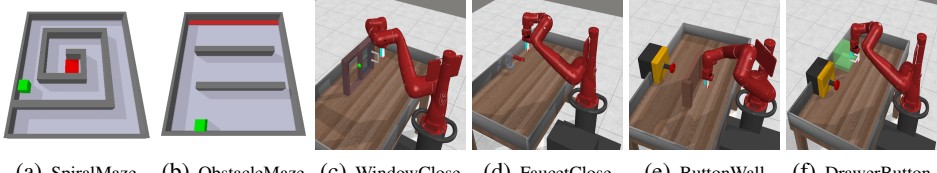

(a) SpiralMaze  (b) ObstacleMaze  (c) WindowClose  (d) FaucetClose  (e) ButtonWall  (f) DrawerButton

Figure 2: Evaluation environments adapted from meta-world robotics benchmark.

Table 1: Success rates and std. deviations (%) on test cases with unseen object variations (except *SpiralMaze*).

| METHOD | BC | BC ($\mathcal{D}^*$) | TD3-BC | IQL | MPPI | MBOP | IRIS | IRIS (MULTI-STEP) | VELAP |
|---|---|---|---|---|---|---|---|---|---|
| SPIRAL MAZE | $0 \pm 0$ | $0 \pm 0$ | $0 \pm 0$ | $0 \pm 0$ | $0 \pm 0$ | $0 \pm 0$ | $0 \pm 0$ | $15 \pm 31$ | $\mathbf{94 \pm 3}$ |
| OBSTACLE MAZE | $0 \pm 0$ | $15 \pm 6$ | $35 \pm 22$ | $6 \pm 3$ | $83 \pm 11$ | $40 \pm 25$ | $50 \pm 25$ | $62 \pm 14$ | $\mathbf{97 \pm 2}$ |
| WINDOW | $0 \pm 0$ | $34 \pm 11$ | $16 \pm 8$ | $9 \pm 4$ | $70 \pm 7$ | $23 \pm 4$ | $69 \pm 3$ | $43 \pm 20$ | $\mathbf{78 \pm 4}$ |
| FAUCET | $0 \pm 0$ | $36 \pm 6$ | $13 \pm 7$ | $8 \pm 4$ | $41 \pm 7$ | $33 \pm 2$ | $10 \pm 2$ | $3 \pm 1$ | $\mathbf{51 \pm 12}$ |
| BUTTONWALL | $0 \pm 0$ | $0 \pm 0$ | $2 \pm 2$ | $0 \pm 0$ | $9 \pm 10$ | $0 \pm 0$ | $35 \pm 5$ | $8 \pm 8$ | $\mathbf{76 \pm 9}$ |
| DRAWERBUTTON | $0 \pm 0$ | $0 \pm 0$ | $0 \pm 0$ | $1 \pm 0$ | $0 \pm 0$ | $0 \pm 0$ | $5 \pm 3$ | $0 \pm 0$ | $\mathbf{11 \pm 3}$ |

which a worker policy must achieve. We also examine **IRIS (multi-step)**, where the set of subgoals is generated by shooting a future state sequence using the cVAE. To establish a fair comparison and disentangle the effects of the representation and planner, we use the same representations, dynamics models across all methods. Further details are provided in App. D.

**Tasks**  We consider the simulated visuomotor control tasks depicted in Fig. 2. In *SpiralMaze*, the velocity of a block robot is controlled in order to navigate from the outermost point of a maze to the inner region. This task was designed to require farsighted planning, as the temporal distance to the goal is approximately 150 steps. In the *ObstacleMaze* environment, the block robot must travel to the opposite wall of the room while two randomly positioned obstacles appear in the center of the workspace. Additionally, we evaluate the *WindowClose* and *FaucetClose* environments from the meta-world [18] benchmark. As done in [6], we use sparse binary rewards and render images from a static camera. Moreover, we propose two new settings *ButtonWall* and *DrawerButton*. In *ButtonWall*, the robot must first navigate around a wall (varying position) and then press a button. To solve the task *DrawerButton*, the agent must first close a drawer and then press a button (both randomly initialized). Our training data $\mathcal{D}$ consists of random trajectory data and a small number of noisy expert demonstrations. We use a latent space of size 32 and RGB images of resolution $64 \times 64$. For details on the environments, data collection and hyperparameters see App. B.

**Results**  An overview of the numerical benchmark evaluation is given in Table 1. VELAP consistently outperform the baselines across all environments in terms of average episode success rate. The improvements are particularly visible in tasks which require far-sighted planning such as *SpiralMaze* and *ButtonWall*. These results support that our tree-based exploration strategy is indeed effective at planning for sparse-reward offline RL. Fig. 3 illustrates a 2D embedding of a latent path for the *SpiralMaze* task computed with VELAP. It suggests that our method explores global solutions and identifies one which reaches through the entire maze. Fig. 3 demonstrates similar capabilities for the *ObstacleEnv* task. Further, it supports that our representation accommodates for the different locations of obstacles, creating latent spaces that mirror the topology of the underlying state space. Ablation results concerning the embedding and dynamics model are provided in App. E.

## 5.2 Physical robot experiments

To test our method under realistic conditions, we designed two manipulation tasks using a low-cost robot arm (Fig. 4). In the first, the robot must push a sponge into the marked region. In the second task, the robot holds a rope which it must unravel from a cylindrical object. Then it needs to position the held end of the rope precisely onto a designated colored area. The robot is controlled by providing desired end effector position and wrist orientation displacements, resulting in a 4-dimensional action space. Both tasks are challenging in terms of perception and control as

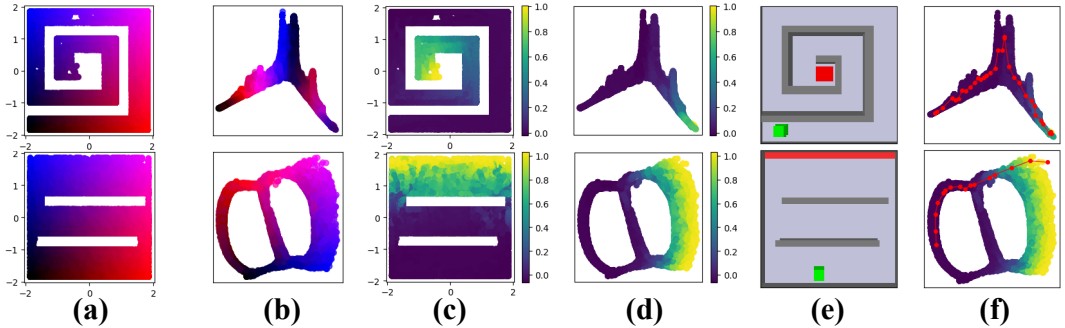

Figure 3: Supporting visualizations for *SpiralMaze* (top row) and *ObstacleMaze* (bottom row) tasks. (a) x-y robot positions for a uniformly sampled set of states (b). 2D Isomap embeddings of learned latent space (color encodes correspondence to robot positions in (a)).(c) approximated global Q-values for x-y robot positions. (d) approximated global Q-values for Isomap embedding of learned latent state space. (e) example image input frames. (f) planned latent paths (red) computed with VELAP and projected to Isomap embedding.

the initial configuration of the objects are randomized. The results of a comparison with BC, BC ($\mathcal{D}^*$) and IRIS are presented in Table 11 (more details in App. E.1). Our method surpasses the performance of its baselines achieving $14/20$ successful episode on the sponge manipulation and $8/20$ on the rope manipulation task. Video demonstrations can be found at https://sites.google.com/view/velap-corl/home.

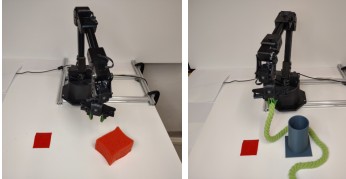

Figure 4: Physical manipulation of a sponge (left) and rope (right).

## 6 Limitations and Future Directions

Our method lays the groundwork for future enhancements. While VELAP was designed for offline RL, it can be adapted to the online setting by interleaving data collecting and learning [3, 6]. It could also enhance sample efficiency by improving policy and critic updates through planning. [56, 4, 31]. Currently, VELAP uses deterministic dynamics and encoder models, limiting it to fully-observable MDPs. By incorporating probabilistic transition models and state filtering approaches (similar to [3, 57]), it can be extended to partially observable and stochastic settings. While our method outperforms existing baselines, it still struggles with complex tasks which we account to the remaining difficulty in estimating accurate latent dynamics for long-horizon planning. Presently, our planner selects paths by minimizing the distance to high-valued states. To further minimize the effects of model inaccuracies, the planning strategy could incorporate uncertainty propagation and assessment in the tree branches. Notably, we discovered that in more complex tasks, such as *DrawerButton* and the rope manipulation scenario, failures often occurred when the agent moved to the final goal region without completing the required subtask, like first closing the drawer or maneuvering the rope around the pole. We believe that improved handling of uncertainties, along with risk-aware measures, could potentially lessen the planner's greedy exploitation of model errors. In this context, planning in belief spaces [58] provides another potential improvement avenue.

## 7 Conclusion

We introduced VELAP, an agent designed for model-based planning in sparse reward offline RL. Diverging from the usual model-based RL planners, our approach employs a tree-based exploration algorithm inspired by sampling-based planners commonly used in robot motion planning. Through empirical evaluation on visual control tasks, we showcased substantial enhancements achieved by our method compared to existing benchmarks. Our aim is that these outcomes will inspire additional exploration into the fusion of sampling-based planning, representation learning, and model-based RL

**Acknowledgments**

This work was partially supported by the Wallenberg AI, Autonomous Systems and Software Program (WASP) funded by the Knut and Alice Wallenberg Foundation.

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

# A   Additional material

Code examples and additional material will be uploaded here https://sites.google.com/view/velap-corl/home. All models were implemented in Python using the PyTorch library. The total training time including all models and baselines amounts to approximately 150 hours (wall clock) on a single GPU.

# B   Hyperparameters and algorithm details

Here we present a description of the hyperparameters of our trained models and planning module.

## B.1   Model architectures

Table 2: Hyperparameters of the encoder $\phi$

| Parameter | Value |
|---|---|
| Batch-norm. | yes |
| Filters | [32,32,64,64] |
| Kernels | [4,4,4,4] |
| Strides | [2,2,2,2] |
| Activation | LeakyRelu |
| Dense layers | [256, 128, 32] |

Table 3: Hyperparameters of dynamics model $h$

| Parameter | Value |
|---|---|
| Batch-norm. | yes |
| Activation | LeakyRelu |
| Dense layers | [128,128,128,128,32] |

Table 4: Hyperparameters of action sampler $g$ ($\beta$-VAE)

| Parameter | Value |
|---|---|
| Batch-norm. | yes |
| Activation | LeakyRelu |
| Latent dimension | 16 |
| $\beta$ (kl-weight) | 0.01 |
| Encoder dense layers | [128,128,128, 2*16] |
| Decoder dense layers | [128,128,128, $d_{\text{action}}$,] |

Table 5: Hyperparameters of policy networks $\pi^l$ and $\pi^g$

| Parameter | Value |
|---|---|
| Batch-norm. | yes |
| Activation | LeakyRelu |
| Dense layers | [128,128,128, $d_{\text{action}}$] |

Table 6: Hyperparameters of critic networks $Q_k^l$ and $Q^g$

| Parameter | Value |
|---|---|
| Batch-norm. | yes |
| Activation | LeakyRelu |
| Dense layers | [128,128,128, 1] |

## B.2 Training hyperparameters

Table 7: Model training hyperparameters

| Parameter | Value |
|---|---|
| batch size | 64 |
| learning rate | 0.0003 |
| $c_0$ | 0.2 |
| $c_1$ (*SpiralMaze*) | 0.001 |
| $c_1$ (*ObstacleMaze*) | 0.01 |
| $c_1$ (metaworld tasks) | 0.01 |
| $c_2$ | 0.001 |
| $c_3$ | 0.001 |
| $c_3$ (expert) | 0.5 |
| $\gamma$ (discount factor) | 0.96 |
| $d_{\mathcal{Z}}$ | 32 |
| $T$ (temperature) | 1.0 |
| $n_{\text{ens}}$ | 3 |

## B.3 Planner and controller hyperparameters

Table 8: Hyperparameters of planner

| Name | Description | Value |
|---|---|---|
| $n_{\text{iter}}$ | Number of planner iterations | 250 (500 in *ButtonWall*) |
| $n_{\text{sim}}$ | Number of simulation steps during tree expansion | 5 (10 in *SpiralMaze*, *ButtonWall* and *DrawerButton*) |
| $\tau_{\text{discard}}^{\text{high}}$ | Q-value threshold for discarding node if too close to existing nodes in the tree | $\gamma^2$ |
| $\tau_{\text{discard}}^{\text{low}}$ | Q-value threshold for discarding node if too far from expansion node | $\gamma^{n_{\text{sim}}}$ |
| $\tau_{\text{discard}}^{\text{std}}$ | Q-value threshold for discarding node if standard deviation of ensemble prediction is too high | $1.0 - \gamma$ |
| $\tau_{\text{neigh}}$ | Q-value threshold to determine neighboring nodes | $\gamma^3$ |
| $\tau_{\text{goal}}$ | Q-value threshold to determine goal nodes | $\gamma^5$ |
| $d_{\text{neigh}}$ | Euclidean distance threshold to determine candidate neighbors | 3 x upper 5-percentile of Eucl. distances between encoding of subsequent states |

Table 9: Hyperparameters of controller

| Parameter | Description | Value |
|---|---|---|
| $n_{\text{replan}}$ | Planning frequency | 15 (25 in *SpiralMaze*, *ButtonWall*) |
| $\tau_{\text{stop}}$ | Q-threshold to stop planning when close to the goal | $\gamma^5$ |
| $\tau_{\text{wp}}$ | Q-threshold for switching to the next waypoint | $\gamma^3$ |

## B.4 Training of policy and value functions

We use TD3-BC [49] as the base offline RL algorithm to train our local and goal policies $\pi^l$ and $\pi^g$, respectively state-action value functions a $Q_k^l$ and $Q^g$. Within our planning framework $Q_k^l$ takes an important role as it provides us with a distance proxy. To further improve the accuracy of $Q_k^l$, we use $n_{ens}$ Q-networks (instead of 2 usually used in TD3). During the training update of the Q-network, we then determine the Q-target by taking the minimum value among the predictions given by the ensemble of Q-networks (similar to [50]). The ensemble further allows us to filter out unlikely or out-of-distribution transitions generated during the tree expansion. This is done by assessing the minimum predicted ensemble Q-value and the standard deviation among the predicted values (Sec. 4).

Our models $\pi^l$ and $Q_k^l$ describe goal-reaching policy and state-action value functions which require a set of goal-conditioned reaching experiences for training. Since our original dataset $\mathcal{D}$ might not describe this particular setting, we can augment it using hindsight goal relabeling. In particular, we create a new dataset $\mathcal{D}'$ consisting of transitions $(z_t, a_t, r_t, z_{t+1}, z_*, \gamma) \in \mathcal{D}'$ by relabeling the values of $r_t$, $\gamma$ ($\gamma$ also indicates terminal condition, i.e. $\gamma = 0$) and adding a goal state $z_*$. We apply a combination of three different relabeling strategies (a) set goal $z_*$ to be next state in the relabeled transition and set $\gamma = 0$; $r_t = 1$ (b) sample $z_*$ from the set of future states within the same trajectory and set $r_t = 0$ (c) sample $z_*$ from another trajectory in the data and set $r_t = 0$.

## B.5 Training of dynamics model

Our dynamics model $h$ is trained using the InfoNCE contrastive loss introduced in [53]. Given an initial state $z_t$ and a sequence of actions $a_{t:t+k-1}$, we use $h$ to generate predictions $\widetilde{z}_{t+k}$ for $i=1..k$. We compute the NCE loss at each each $k$ with positive pairs $(\widetilde{z}_{t+k}, z_{t+k})$ and take negative examples $z_j$ randomly from the batch. We use $f = e^{-||z_i - z_j||_2/T}$ to compute similarity between latent encodings. Our overall training loss for $h$ (Eq. 9) takes the average over the contrastive loss terms computed at all $k$ steps. For all experiments, we use $k = 3$.

$$\mathcal{L}_h = -\frac{1}{k} \sum_k \mathbb{E}_{\mathcal{D}} \Big[ \log \frac{f(\widetilde{z}_{t+k}, z_{t+k})}{\sum_j f(\widetilde{z}_{t+k}, z_j)} \Big] \tag{9}$$

## B.6 Additional details about planning method

**Neighbor computation** To determine if a newly sampled node $z_{new}$ is novel, we check its similarity to existing nodes in the tree by evaluating the state-action value function. Computing the goal-conditioned values with respect to all nodes results in an enormous computations overhead. Yet, we can significantly reduce the amount of computation by first determining a set of candidate neighbors around $z_{new}$ using the Euclidean metric and a distance threshold $d_{neigh}$. In practice, we found it useful to define $d_{neigh}$ based on the statistics of Euclidean distances between subsequent states in the dataset (see App. B.3).

**Batch processing** The method in Alg. 1 describes an iterative schema for which at every expansion step one new node is generated and evaluated. Yet, some steps can be computed in parallel on a GPU in order to speed up the planning time. For a practical implementation, we therefore suggest to parallelize the tree expansion by sampling multiple expansion nodes at once and generating new nodes by passing batches through the neural network dynamics model. Similarly, we can compute state-action values in batches instead of assessing one new nodes at a time. For discussions about highly-parallelized implementations of classical RRT-like planners, we refer to [59, 60].

## B.7 Additional details about MPC controller

Alg. 2 provided below outlines the pseudocode for our MPC controller. The function `update_waypoint`($z_{curr}, g^*$) is responsible for determining the subsequent waypoint $z_{wp}$ that our

local policy aims to attain. Specifically, we estimate the value between the current state and waypoint and switch to the next element in $g^*$ if the predicted value surpasses a threshold $\tau_{\text{wp}}$, i.e $Q_{\min}^{\text{curr,wp}} > \tau_{\text{wp}}$. As we approach the final goal, indicated by the proximity of our current state $z_{\text{curr}}$, we stop the planning and compute actions based on the policy $\pi^g$. To ascertain our proximity to the goal, we compare the predicted value of the global value function $Q^g$ against a predefined threshold as $\tau_{\text{goal}}$.

---

**Algorithm 2** MPC controller

---

Given: $s_{\text{init}}$, $n_{\text{replan}}$ $n_{\text{max\_steps}}$, $\phi$, $\pi^l$, $\pi^g$
$z_{\text{curr}} \leftarrow \phi(s_{\text{init}})$            ▷ Map state to latent encoding
$i \leftarrow 0$
**while** goal not achieved and $i < n_{\text{max\_steps}}$ **do**
    **if** not $i$ mod $n_{\text{replan}}$ **then**          ▷ Replan every $n_{\text{replan}}$ steps
        Build tree $\mathcal{T}$ rooted at $z_{\text{curr}}$ for $n_{\text{iter}}$ steps (Alg. 1).
        Determine $g^* = \{z_{curr}, z_1, .., z_n\}$ given $\mathcal{T}$
    **end if**
    $z_{\text{wp}} \leftarrow \texttt{update\_waypoint}(z_{\text{curr}}, g^*)$      ▷ Update waypoint if close enough
    $a \leftarrow \pi^l(z_{\text{curr}}, z_{\text{wp}})$     (or use $\pi^g$ within proximity to the overall goal)     ▷ Compute next action
    Execute $a$ and observe updated state $s_{\text{curr}}$
    $z_{\text{curr}} \leftarrow \phi(s_{\text{curr}})$
    $i \leftarrow i + 1$
**end while**

---

# C    Evaluation Environments

## C.1    Description of block environments

Similar to the evaluation environments in [15], we implement two long-horizon navigation tasks characterized by a comparatively low-dimensional underlying state space. This aspect facilitates a visual examination of the learned latent embeddings via dimensionality reduction methodologies such as Isomap [61]. Within both scenarios, a block-shaped robot's motion is controlled through velocity commands, while its movement remains confined to a two-dimensional plane.

### C.1.1    SpiralMaze

To accomplish this objective, the block agent is tasked with maneuvering from the outer edge of a spiral-shaped corridor to the inner area colored in red (Fig. 2a). The episode's duration is capped at 300 steps. During data generation for training, the agent is initialized at collision-free positions within the workspace. Subsequently, random action sequences are executed by sequentially adding Gaussian noise to an initially sampled uniformly random action at the start of each episode. For testing purposes, the agent's position is uniformly sampled from a small area in proximity to the outermost point of the spiral-shaped corridor.

### C.1.2    ObstacleMaze

Within this setting, the agent is required to move towards the upper wall of the workspace, highlighted in red (Fig. 2b). To successfully accomplish this objective, the agent needs to execute actions that navigate around two obstacles, positioned randomly near the workspace center at the beginning of each new episode. The maximum permissible number of steps in the environment is 100. During testing, the agent's initial configuration is randomly set in close proximity to the wall opposite to the goal. The same data collection policy as for the *SpiralMaze* task was utilized.

## C.2    Description of manipulation environments

We have customized several environments of the meta-world robot benchmark tasks proposed by [18]. The underyling physics simulation engine is Mujoco, as introduced by [62]. To enable vi-

sual manipulation, similar to the problems studied in [6], we enable background rendering of RGB images from a static viewpoint. The robot is controlled by commanding desired end effector and gripper opening displacements resulting in a 4-dimensional action space. While *WindowClose* and *FaucetClose* were with small modifications adapted from [18], we evaluate two new environments *ButtonWall* and *DrawerButton*. These new scenarios were purposely designed to investigate tasks with long horizons under sparse rewards. Importantly, they require the integration ("stitching") of trajectory data originating from different regions of the workspace to determine viable solutions.

In our data collection process, we employ a suboptimal policy that predominantly applies random actions (using additive Gaussian noise). Infrequently, this policy selects actions from a pre-defined expert policy. Table 10 provides insight about the number of samples and trajectories in the training data and presents the portion of successful transitions (reward=1). Across all manipulation tasks, we set the maximum permitted number of environment steps at 150, with the exception of the *Button-Wall* scenario, where we allow up to 250 steps during the evaluation phase.

### C.2.1  WindowClose

In order to accomplish this task, the robotic arm must successfully open a window by shifting a specific handle sideways. We implement environmental variability by randomly determining the x-y location of the window object in each episode. During the data collection stage, we randomly set the positioning of the end effector above the surface of the table. The sampling of expert actions is restricted to areas close to the goal region (window handle). This approach is intended to guarantee that the strategy employed necessitates to "stitch" different trajectories together to reach the objective and complete the task when starting from states that are farther away. To ensure challenging planning situations during testing, we initiate the robot at a significant distance away from the target.

### C.2.2  FaucetClose

This task is similar to the *WindowClose* task, but it requires the agent to use its end effector to close a faucet instead. In addition, we employ analogous strategies for data gathering and scenario creation as those used in the *WindowClose* environment.

### C.2.3  ButtonWall

In this scenario, the robot's end effector is required to navigate around a wall structure before pressing a button. The location of the wall is randomly set at the beginning of each episode. Furthermore, a height limitation is imposed on the end effector to ensure that the agent takes a more extended path around the wall, as opposed to simply elevating the end effector. The dataset was produced by either placing the agent in front of the wall, near the button, or far behind the wall. However, expert samples in the dataset only exist for scenarios when starting closer to the goal. For testing purposes, the end effector is sampled within a restricted region behind the wall.

### C.2.4  DrawerButton

In this scenario, the agent is tasked to first close a drawer using its end effector and subsequently press a button. To train the agent, we develop a dataset by separately collecting trajectories for each subtask. This approach necessitates the use of a method capable of combining different trajectories in the data to devise a solution that achieves the overall task goal.

### C.3  Composition of training dataset

The table below presents the composition of our training datasets. Each context refers to a new environment initialization (excl. agent) such as the position of obstacles.

Table 10: Composition of training datasets for each environment

| Environment | Num. contexts | Traj. per context | Max. traj. length | Successful transitions |
|---|---|---|---|---|
| *SpiralMaze* | 1 | 1000 | 20 | 0.12 % |
| *ObstacleMaze* | 250 | 20 | 20 | 0.11 % |
| *WindowClose* | 200 | 10 | 50 | 0.48 % |
| *ButtonWall* | 200 | 10 | 50 | 0.16 % |
| *FaucetClose* | 200 | 10 | 50 | 0.31 % |
| *DrawerButton* | 150 | 20 | 50 | 0.16 % |

## D  Baseline methods

To enable a fair comparison between different methods, we use the same underlying representation/encoder $\phi$ and dynamics model $h$ in the evaluation of all baselines. For assessing the quality and impact of our representation learner, please review the experimental ablation study in App. E.2.

### D.1  BC and BC ($\mathcal{D}^*$)

Simple behavioral cloning baselines for which we use the same network architecture as our policy networks (see Table 5) and train using a mean-squared error objective on the predicted actions. For $\mathcal{D}^*$ we train only on the subset of successful episodes in the dataset. For each method, we train for $3 \cdot 10^5$ iterations using a learning rate of $3 \cdot 10^{-4}$ and batches of size 128.

### D.2  TD3-BC [49]

This baselines resembles the underlying global policy $\pi^g$ used in VELAP. It provides us with a baseline to assess how well pure offline RL performs without any additional planning methods.

### D.3  IQL [55]

This method presents a state of the art model-free offline RL baseline which utilizes expectile regression to estimate state-conditional expectiles of the target values in order to avoid querying values of out-of-distribution actions during training. We train IQL for $3 \cdot 10^5$ iterations with a learning rate of $3 \cdot 10^{-4}$, batches of size 256 and using hyperparameters $\tau = 0.7$ and $\beta = 3.0$.

### D.4  MPPI

We implement a trajectory optimization baseline similar to the model-based planning algorithm introduced in [7]. The method in [7] presents an adaption of MPPI specifically for the online reinforcement learning setting which optimizes the expected return of sampled trajectories. To estimate the return, a learned model is used to predict the reward for each trajectory node while a learned Q-function predicts the future return beyond the specified planning horizon. Since rewards in our evaluation environments are sparse, the predicted rewards carry little guidance for the trajectory optimization as most states have 0 reward. Therefore, we adapt the objective in [7] and instead use the accumulated sum of state-action values as the optimization criterion. This type of scoring function in model-based RL has recently been discussed in [63]. To implement this baseline, we utilize the Q-function of TD3-BC. For all environments, we use 1000 samples per iteration, a planning horizon of 50, elite size 64 and 5 iterations. Replanning is done every 5 environment steps.

### D.5  MBOP [36]

MBOP presents an adaptation of MPPI which was particularly designed for the offline RL setting. It generates new candidate trajectories by addding a small amount of Gaussian noise to the actions predicted by a behavioral-cloned policy. To evaluate the quality of the rollouts it uses a truncated value function trained on the offline data. Due to the sparse nature of rewards in our experiments, we

found that both the behavioral-cloned policies and the truncated value function were insufficient to generated farsighted behaviors that solve our tasks. To accommodate for the long planning horizons, we instead sample actions from our TD3-BC policy and use the corresponding Q-values to assess candidate trajectories during the optimization (similar to our MPPI baseline). For all environments, we use 1000 samples per iteration, a planning horizon of 50, elite size 64, 5 iterations and $\beta = 0.7$. Replanning is done every 5 environment steps.

### D.6  IRIS [37]

### D.7  IRIS (multi-step)

We evaluate an extension of IRIS in which we use the state prediction model (conditional VAE) to generate multi-step rollouts of suitable subgoals. This strategy increases the exploration horizon and allows to choose the best subgoal from a larger and potentially more diverse set of states. This planning strategy can also be seen as random shooting of coarse subgoal sequences. In all experiments, we generate 256 different trajectories using rollouts of length 5 and a conditional generative model to predict states for a horizon of 5. In our evaluation, we found that this method sometimes performs worse than IRIS. We attribute this to the fact that the global policy doesn't align with the capabilities of the local one, which occasionally results in the selection of subgoal states that might not be attainable.

## E  Supplementary Experiments and Analysis

### E.1  Physical hardware experiments

The physical evaluation was performed using the WidowX 200 low-cost robot platform. For the real-world validation of our method, we collected 200 episodes of data for the sponge ($\sim 15000$ samples) and 150 episodes of data for the rope manipulation ($\sim 15000$ samples) tasks. Training data was generated by operating the robot through a gamepad and took less than 1 hour per task. The collected dataset consist largely of suboptimal and entirely random trajectories. Successful transitions (positive reward + episode termination) were labeled manually during the data collection. Similar to our simulated datasets, we collect trajectories in such a way that successful episodes always start within the vicinity of the goal. Conversely, during testing, we deliberately position the agent distant from the goal region. Consequently, this configuration emphasizes the need for an approach capable of internally assessing the connectivity between distinct trajectory segments within the data. To construct states, we combine three sequential images captured by a stationary camera. The results of a comparison with BC, BC ($\mathcal{D}^*$) and IRIS are presented in Table 11.

Table 11: Results of phyiscal robot experiments (successful episodes)

| Environment | BC | BC ($\mathcal{D}^*$) | IRIS | VELAP |
|---|---|---|---|---|
| Sponge | 5/20 | 6/20 | 6/20 | 14/20 |
| Rope | 0/20 | 0/20 | 2/20 | 8/20 |

**Sponge task**  In this setting, the robot needs to push a sponge object onto a marked goal region (Fig. 4 (left)). To increase the difficulty of this task, we initialize the robot end effector between the goal region and the sponge. Consequently, the robot must initially maneuver itself behind the sponge before it can proceed to push the sponge towards the goal. During both training and testing, the initial poses of the end effector and sponge object were subject to random selection.

**Rope task**  In this particular setting, the robot's end effector grasps a green rope, requiring the robot to skillfully maneuver the rope around a central pole in the workspace before ultimately placing the held end within a designated goal area (as shown in Figure 4 (right)). To elevate the complexity of this scenario, the agent is required to first unwind the rope from the pole before moving towards the

goal region. To introduce a challenging long-horizon aspect, we initialize the end effector in close proximity to the goal region while the rope is partially wrapped around the pole.

## E.2 Ablating the impact of the learned representation

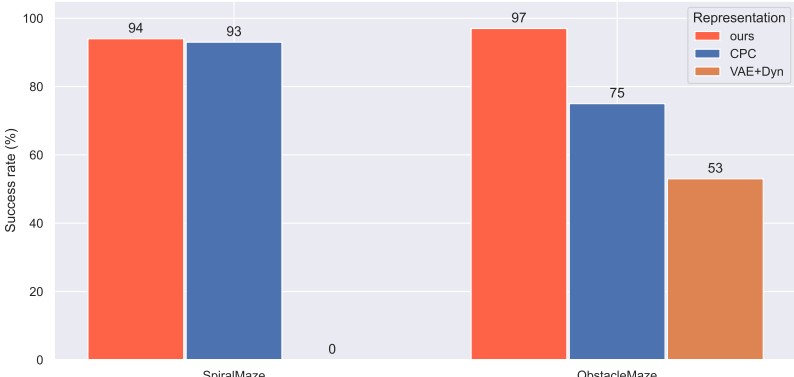
Figure 5: Impact of the type of representation on the performance of our method.

## E.3 Influence of the dynamics loss

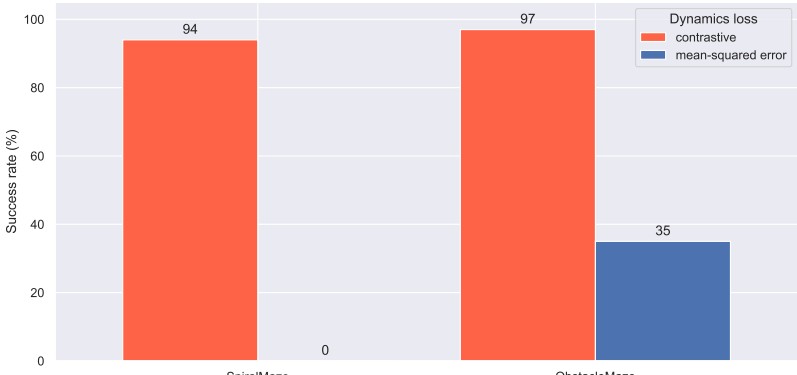
Figure 6: Impact of the type of dynamics loss on the performance of our method.

## E.4   Computation time

Here, we provide an assessment of the computation time needed and the resulting success rates achieved by our approach in comparison to MPPI. Both algorithms were tested on hardware featuring a NVIDIA GeForce RTX 3090 graphics card. Despite utilizing GPU computation and implementing both methods in PyTorch, we didn't specifically fine-tune either for computational speed. It is worth mentioning that MPPI recalculates its plans every 5 steps within the environment, whereas our method follows a 25-step interval in the *SpiralMaze* and 15 steps in the *ObstacleMaze*.

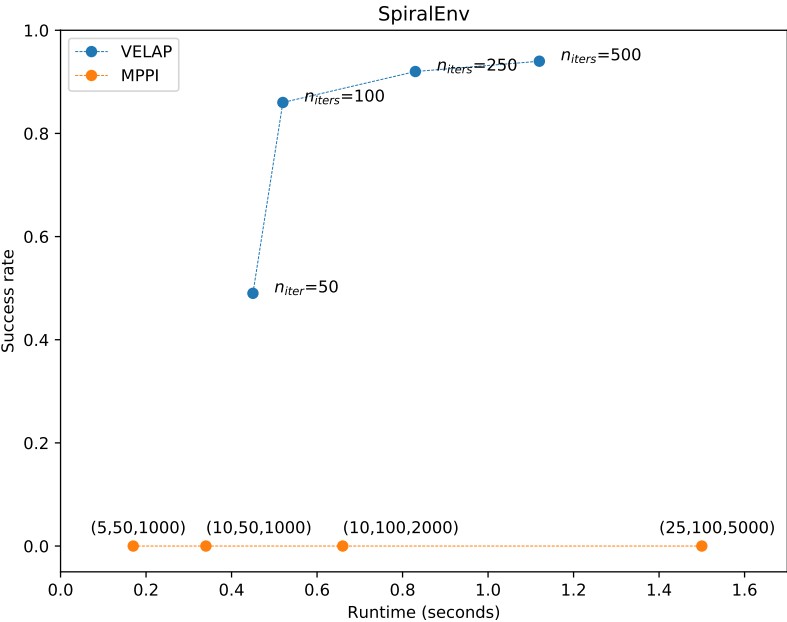

Figure 7: Relationship between average episode success rates and single planning query runtime (test scenarios) for different planning hyperparameters on *SpiralMaze* environment. For MPPI, we report results for varying (iterations, horizon, number of samples).

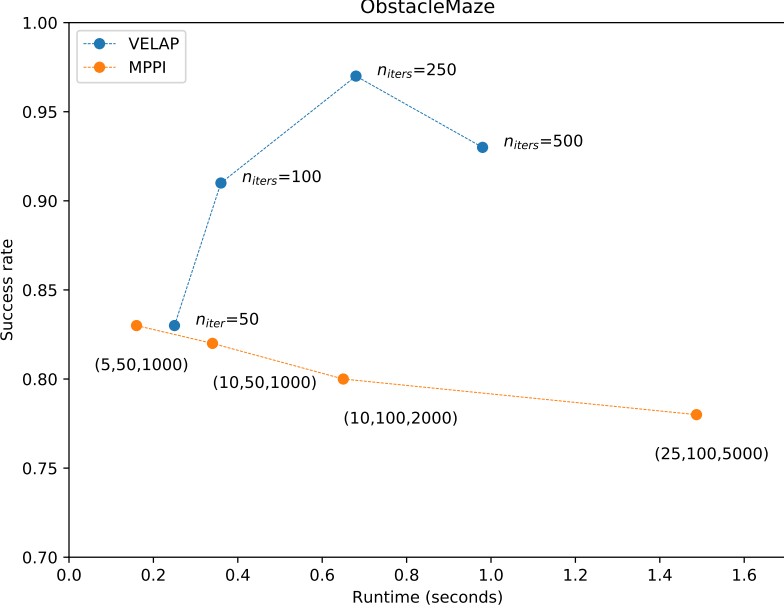

Figure 8: Relationship between average episode success rates and single planning query runtime (test scenarios) for different planning hyperparameters on *ObstacleMaze* environment. For MPPI, we report results for varying (iterations, horizon, number of samples).

