# OpenReview forum: "Expansive Latent Planning for Sparse Reward Offline Reinforcement Learning"
_robot-learning.org/CoRL/2023/Conference — CoRL 2023 Oral_

### Official Review · Reviewer_foWD · 2023-07-09

**Confidence:** 4
**Originality:** Excellent
**Technical Quality:** Excellent
**Clarity Of Presentation:** Good
**Impact:** 4

**Recommendation:**

Strong Accept: I recommend accepting the paper and will argue for my recommendation even if other reviewers hold a different opinion.

**Review:**

The proposed algorithm is quite complex, consisting of many components, but the overall architecture makes sense.
The offline reinforcement learning setting is a good choice as the static data set allows to evaluate the proposed planner better.
Experiments demonstrate convincing performance on challenging tasks relying only on image observations.
It is a promising approach that would be very interesting to be evaluated in an online reinforcement learning where the problem of exploration for data collection needs to be addressed.



**Quality Of The Limitations Section:**

Limitations are addressed clearly

**Questions For Rebuttal:**

Issues:
1) After stating in line 287 that the simulated datasets consist primarily of random trajectories. This is misleading since the reader assumes the same thing would be true for the experiments on the real robot. Only in the appendix is it mentioned that the physical robot datasets consist only of expert demonstrations collected with a joystick.
2) line 307: It is unclear how the position and orientation of the end effector together only have 4 DoF. Adding that you only control the z-orientation of the end effector takes nearly no space.
3) The type of physical robot should be stated somewhere. At least in the appendix.

Questions:
- In Table 1, why do the success rates come with standard deviations?
- How did you choose the dimensionality of the latent state?

Suggestions:
- In the physical robot experiments section, it is not clear to the reader that the robot needs to unwind the rope from the pole. Without this information, at first glance, the task seems like a simple reaching task where the rope does matter.
- Despite the otherwise high quality of the paper, the submitted version still has a considerable number of spelling mistakes.

**Robotics Focus:**

Sufficient demonstration on hardware

**Summary Of Paper:**

The authors present a novel model-based planner that utilizes tree-based exploration guided by a policy and Q-function ensemble trained with TD3-BC in an offline reinforcement learning scenario on a dataset consisting of expert demonstrations. For simulated environments the datasets were augmented with random trajectories that were relabeled in hindsight. Rollouts are predicted in a latent space using a transition model that was trained jointly with a variational autoencoder on pure image observations.
While the approach is not limited to the offline reinforcement learning setup, the static dataset allows to share the learned components such as policy, value function, auto-encoder, transition model, etc to fairly compare their novel planner against baseline planners.
The proposed approach outperforms previous approaches in simulated settings requiring long horizon planning and successful rollouts on a physical robot are demonstrated.

**Summary Of Recommendation:**

I suggest accepting this work.

I have listed some presentational issues and suggestions. After addressing issue 1 (my main concern), I will suggest strong accept.

---

### Official Review · Reviewer_xuPo · 2023-07-16

**Confidence:** 4
**Originality:** Good
**Technical Quality:** Good
**Clarity Of Presentation:** Very Good
**Impact:** 3

**Recommendation:**

Weak Accept: I recommend accepting the paper, but will not argue for my recommendation if the majority of other reviewers have a different opinion.

**Review:**

Strength:
* The method is novel. Although there has been previous work in combining sampling-based motion planning with RL, this instantiation is novel and well justified.
* The experiment results are strong and comprehensive, including multiple tasks, real robot experiments, and comparisons against multiple baselines.

Weaknesses:
* Missing baseline of shooting+Q-function: The introduction mentioned that “Even with guidance from value heuristics, such as the one proposed in [7], local minima may still impede progress, particularly when estimating the optimal value function is difficult due to sparse reward feedback or limited training data”. However, this claim is not supported in the experiments. It would be great to have a head-to-head comparison between shooting and the proposed planning component given the same Q-function. Another related work: “Learning Off-Policy with Online Planning” from corl 2021.
* Lack of ablations: The proposed method has many components. It is important to understand the importance of different components.

Other comments:
* It’s hard to understand Figure 3 and 4.
* What’s the cost of computation for this method?
* The “action model” in Eqn (1) is referred to as “a conditional generative model” in line 172. It would be more clear to also include the word “action model” in the text. Also, R^m seems to be a random number for the generative model. Please clarify in the text.
* Line 170 typo: memeber
* “Alignment of representation and planner”: The flow of this paragraph is a bit confusing. As far as I understand, the authors are trying to say two things: First, we need state representation that favors long-horizon dynamics. This is achieved by CPC. Second, we need state representation that matters for control. This is achieved by joint optimization of the dynamics and policies with a shared encoder. It would be great to mention these two considerations more separately.
* “To help our planner discover task-relevant areas quicker, we further predict actions with pi^g probability p^policy” What does it mean?

**Quality Of The Limitations Section:**

Limitations are addressed clearly

**Questions For Rebuttal:**

Please address the weaknesses mentioned above.

**Robotics Focus:**

Sufficient demonstration on hardware

**Summary Of Paper:**

The paper presents a method that combines sampling-based motion planning with reinforcement learning to tackle sparse reward offline RL problems. The planning component of the proposed method operates on the embedding space of the states. The Q-function given from an offline RL algorithm (TD3-BC) is used to guide planning. The experiments demonstrate the effectiveness of the proposed method across multiple tasks, including both sim and real tasks.

**Summary Of Recommendation:**

The paper proposes a novel method of combining sampling-based planning and RL. It shows strong empirical results with real robot experiments. The paper can be further improved by including more ablations and a fair comparison to shooting+Q-function methods.

---

### Official Review · Reviewer_dVQH · 2023-07-19

**Confidence:** 4
**Originality:** Very Good
**Technical Quality:** Good
**Clarity Of Presentation:** Good
**Impact:** 4

**Recommendation:**

Strong Accept: I recommend accepting the paper and will argue for my recommendation even if other reviewers hold a different opinion.

**Review:**

Compare to other model-based offline RL or offline planning driven RL methods, this work cleverly uses the pretrained offline Q function to guide the planning. To reduce the potential redundancy of the search tree, VELAP propose a Q-value guided tree-expansion algorithm which seems more efficient and can conduct faster planning practically.

**Quality Of The Limitations Section:**

Additional details required

**Questions For Rebuttal:**


Q1 - The current approach employs TD3BC for training the Q functions. It would be beneficial to investigate the impact of using alternative offline Q value estimation methods on performance. How would the results differ if other methods were used? This could provide insights into the strengths and limitations of the proposed approach and contribute to a more thorough understanding of the algorithm's behavior.

Besides, it also would be nice to see, for example, IQL vs VELAP-IQL (q functions trained using IQL style) to see the improvements/harmness.


Q2 - Hyperparameters:
Q2-1: In Equation 2, the weights c0 and c1 are mentioned. To provide a comprehensive evaluation, it is important to disclose the actual values chosen for these weights and elaborate on how they were selected. The choice of weights may significantly impact the planner's performance, so an analysis of their influence on the results should be included.

Q2-2: The paper mentions the hyperparameter n_sim in Algorithm 1, but it lacks information on how the simulation horizon was selected. If the value of n_sim was practically determined based on evaluation experiments, it is recommended to present those experiments and their results in the paper. This would enhance the clarity and reproducibility of the proposed method.

Q3: For "Node sampling and tree expansion", the rejection part is a bit unclear. How does it calculate the value for expanded node? Based on the provided understanding, the Q function estimates the goal-conditioned state-action value. Denoting the father state node as S_f and the expanded node after simulation as S_n, with action a=\pi^g(S_f), VELAP estimates Q^l(S_f, S_n, a) for conducting node rejection. However, this part requires more elaboration to ensure its accuracy and to make the paper more accessible to readers.


Q4: Where is definition of dynamic loss Lh? I read in Appendix and saw MSE vs Contrastive, and got confused about contrastive loss.


**Robotics Focus:**

Sufficient demonstration on hardware

**Summary Of Paper:**

In this study, the authors make a significant advancement towards incorporating sampling-based planning into the reinforcement learning framework, particularly for addressing challenging sparse-reward control tasks with high-dimensional inputs. Their proposed method, VELAP (Variational Exploration of Latent Action Plans), navigates through sequences of waypoints using a state embedding learned from the environment.

Unlike conventional sampling-based techniques, VELAP employs an iterative approach, wherein it expands a tree-based memory that tracks the visited latent areas. This memory is instrumental in exploring a more extensive region of the latent space within a given set of search iterations. By leveraging this memory-based exploration, VELAP aims to achieve more efficient and effective exploration of the environment's state space, enhancing the overall performance in solving tasks with sparse rewards.

**Summary Of Recommendation:**

The idea of using retrained offline Q function to guide planning is novel and interesting. The node sampling and tree expansion part improves the sample-efficiency of planning.

---

### Official Review · Reviewer_6i5h · 2023-07-20

**Confidence:** 4
**Originality:** Good
**Technical Quality:** Very Good
**Clarity Of Presentation:** Poor
**Impact:** 3

**Recommendation:**

Weak Accept: I recommend accepting the paper, but will not argue for my recommendation if the majority of other reviewers have a different opinion.

**Review:**

**Clarity**:
Although the writing itself is of good quality, the presentation is frequently confusing. In many places, information feels scattered, unclear, or missing throughout the paper. For instance, on line 194, it's not clear whether hindsight goal relabelling is used to train both policies, or just one, and which policies are trained this way if so. In equation 5, the first term of the $\mathcal(L)_\pi^l$ equation does not appear to depend on $\pi^l$, which makes it unclear why it appears in this term. On line 221-224, the authors describe two separate cutoffs for the selection of points based on the mean and standard deviation of the Q ensemble, but it's not clear why they don't just calculate the much-more-standard lower confidence bound from these values.  The choice of node sampling probabilities feels arbitrary, as a split between two formulas that seem to come from nowhere and a random uniform distribution that also comes from nowhere. The loss function for the dynamics model h appears in the loss for the model, but not actually shown anywhere. It's not clear why the local Q function updates according to the global Q function instead of following the normal Bellman equation. It's also not clear how the local Q function loss allows for hindsight goal relabelling, as hindsight typically changes the goal of the both the current-step-Q and the next-step-Q. However, in the local loss function, the next-step-Q is the global Q function, so the standard substitution is not possible. Figures 3 and 4 are difficult to understand, especially the meaning of the coloration in $a$ and $b$, which is not explained anywhere.

There are many sections of the paper where important information is absent entirely, and many more where unusual choices are made without adequate explanation.


**Originality**:
The work is original in its choice of performing reinforcement learning over an expansive tree search.


**Quality**:

While the results in the experiments section are impressive, it's not clear how much planning time VELAP was given for these problems. This is significant, as it could make for a very unfair comparison with other offline reinforcement learning methods which must operate within a fixed time budget.

Beyond this concern, I find it difficult to assess the quality of the work, due to difficulty parsing technical details from the text.


**Significance**:
Doing reinforcement learning over a sampling-based planner is an interesting idea, and the experiments show strong results on challenging tasks. I would consider the work to be moderately significant.


**Strengths**
- The experiments show significant performance gains compared to each other studied method.
- The idea is interesting and original

**Weaknesses**
- The paper is very hard to follow. This makes it difficult to assess the other merits and demerits of the work.

**Quality Of The Limitations Section:**

Limitations are addressed clearly

**Questions For Rebuttal:**

The main issue with the work is clarity. A sizeable rewrite may be needed to clarify the approach used and the decisions made by the authors

**Robotics Focus:**

Sufficient demonstration on hardware

**Summary Of Paper:**

The authors propose an offline reinforcement learning algorith based on Expansive Tree Search. This method trains both local and global Q and policy networks, and uses them to construct an expansive tree search in the latent space of an encoder model. They use the Q values to set a sampling distribution over the nodes in the tree, and the policy to expand nodes with new actions. They find that the proposed method outperforms other offline reinforcement learning algorithms on a wide range of sparse-reward environments, including physical robot experiments.

**Summary Of Recommendation:**

Although the work has interesting ideas and achieves good results, the presentation is poor, with key information frequently being unclear or missing. Most relevant among these omissions is how long the proposed method was given to plan during the experiments. Since this is a planning method, having additional time when comparing against non-planning RL methods would be a major bias in favor of the authors' approach. for these reasons, I must recommend rejection.

Edit: After the authors changes during the rebuttal period, I believe the work is now significantly more clear and easy to understand. I now recommend acceptance.

---

### Decision · Program_Chairs · 2023-08-30

**Decision:**

Accept (Oral)

**Comment:**

The authors introduce an offline reinforcement learning approach that incorporates sampling-based planning, addressing challenging sparse-reward control tasks. This technique simultaneously trains local and global Q networks and policy networks. Search is performed in  an encoder model's latent space. Q values determine node sampling distribution, while the policy expands nodes with new actions.
The method excels in sparse-reward settings, surpassing alternative offline reinforcement learning methods, as evidenced by diverse experiments, including physical robot scenarios.

The overall verdict by the reviewers is positive. I am happy to suggest accepting this paper.

Please incorporate the improvements discussed during the review/rebuttal into the final version of the paper.